# Preparation and Characterization of Hydroxylated Recombinant Collagen by Incorporating Proline and Hydroxyproline in Proline-Deficient *Escherichia coli*

**DOI:** 10.3390/bioengineering11100975

**Published:** 2024-09-27

**Authors:** Zhimin Cheng, Bin Hong, Yanmei Li, Jufang Wang

**Affiliations:** 1School of Biology and Biological Engineering, South China University of Technology, Guangzhou 510006, China; 202120149891@mail.scut.edu.cn (Z.C.); 202010108485@mail.scut.edu.cn (B.H.); 202010108442@mail.scut.edu.cn (Y.L.); 2Guangdong Provincial Key Laboratory of Fermentation and Enzyme Engineering, South China University of Technology, Guangzhou 510006, China

**Keywords:** recombinant collagen, hydroxylation rate, incorporation, co-expression, triple helix

## Abstract

Collagen possesses distinctive chemical properties and biological functions due to its unique triple helix structure. However, recombinant collagen expressed in *Escherichia coli* without post-translational modifications such as hydroxylation lacks full function since hydroxylation is considered to be critical to the stability of the collagen triple-helix at body temperature. Here, a proline-deficient *E. coli* strain was constructed and employed to prepare hydroxylated recombinant collagens by incorporating proline (Pro) and hydroxyproline (Hyp) from the culture medium. By controlling the ratio of Pro to Hyp in the culture medium, collagen with different degrees of hydroxylation (0–88%) can be obtained. When the ratio of Pro and Hyp was adjusted to 12:8 mM, the proline hydroxylation rate of recombinant human collagen (rhCol, 55 kDa) ranged from 40–50%, which was also the degree of natural collagen. After proline hydroxylation, both the thermal stability and cell binding of rhCol were significantly enhanced. Notably, when the hydroxylation rate approached that of native human collagen (40–50%), the improvements were most pronounced. Moreover, the cell binding of rhCol with a hydroxylation rate of 43% increased by 29%, and the melting temperature (Tm) rose by 5 °C compared to the non-hydroxylated rhCol. The system achieved a yield of 1.186 g/L of rhCol by batch-fed in a 7 L fermenter. This innovative technology is expected to drive the development and application of collagen-related biomaterials with significant application value in the fields of tissue engineering, regenerative medicine, and biopharmaceuticals.

## 1. Introduction

Collagen, the principal structural protein of the extracellular matrix (ECM), plays an important role in preserving tissue architecture and modulating cellular physiological environments [1,2]. It self-assembles into striated fibrils that support cell growth and confer mechanical resilience to connective tissues [3]. The bioactivity of collagen is evidenced by the recognition of collagen motifs by human cell surface receptors, including integrins and discoidin domain receptors (DDRs), which in turn activate intracellular signaling cascades and facilitate cell adhesion [4,5]. Collagen and its derivatives have been extensively utilized in various industries, including cosmetics, pharmaceuticals, and tissue engineering, due to their low immunogenicity and superior cell adhesion properties [6,7,8].

The human genome encodes a diverse array of collagen proteins, with 28 distinct types identified to date [9]. The molecular architecture of collagen is distinguished by its triple helix configuration, a structural motif in which three polypeptide chains intertwine to create a right-handed, coiled, rope-like structure [10]. This unique tertiary structure is fundamental to collagen’s mechanical properties and biological function. Given the broad diversity of collagen and its multifunctional roles, the collagen composition of each tissue is distinctive. In numerous tissues, mature collagen fibrils are heterotypic and comprised of various collagen types [11]. The primary sequence of collagen is marked by a high frequency of glycine residues, which constitute approximately one-third of the total amino acid composition. The highly repetitive (Gly-X-Y) motif is a hallmark of collagen’s primary structure, with proline residues frequently occupying the X and Y positions. Additionally, proline (Pro) and hydroxyproline (Hyp) account for one-fifteenth of the amino acid sequence. These residues have been verified to be pivotal for the triple helix’s folding, stability, and intermolecular interactions [12]. The collagen α chain, formed during translation, requires post-translational modification, mainly including hydroxylation and glycosylation, which contribute to the thermal stability and mechanical stability of triple helix and assembled collagen, respectively [13,14]. Significantly, hydroxylation is a critical process, given that hydroxyproline is an essential constituent in the formation of collagen fibrils, which interact with cellular receptors and molecular ligands [15,16,17,18].

In general, various methods for collagen production include direct extraction from animal tissues and plant systems, as well as recombinant expression in microorganisms [19]. However, collagen extracted from animals is often associated with immune reactivity and the risk of interspecies pathogen transmission [20]. Since the advent of recombinant production techniques in biotechnology, *E. coli* has been a primary host system for the generation of recombinant proteins [21]. The *E. coli* expression system, with its rapid growth, low nutrient requirements, and ease of large-scale cultivation, has taken advantage to produce recombinant collagen production efficiently. Despite this, the lack of post-translational modifications in prokaryotic systems presents a challenge, particularly the absence of hydroxylation at proline residues, which is essential for collagen function [19]. To address this, co-expression of proline hydroxylase with collagen has been proposed. For example, the co-expression of recombinant human collagen with proline hydroxylase from *Bacillus anthracis* (BaP4H) achieved a hydroxylation rate of 63% [22]. However, the regulation of hydroxylation remains an unsolved issue, as the currently introduced proline hydroxylase has not been able to effectively manipulate the hydroxylation rate of collagen [23,24,25,26,27]. Previous studies have shown that exogenous Hyp supplementation during cell culture promotes its co-translation with intracellular Pro [28]. It allows for the incorporation of hydroxylated proline residues into recombinant collagen, thereby avoiding the requirement for proline hydroxylase. However, previous research has reported that recombinant collagens produced in this manner exhibit a high rate of hydroxylation due to the use of only Hyp [29,30].

In this study, a proline nutritionally auxotrophic *Escherichia coli*, capable of efficiently expressing recombinant proteins, was constructed, and a system for the preparation of hydroxylated recombinant collagen by incorporating Pro and Hyp was established (Figure 1). The system was then scaled up to a fermenter to enhance the production of recombinant collagen. Through this approach, recombinant human collagens (rhCols) with varying degrees of hydroxylation were prepared and characterized for their thermal stability and biological activity for further understanding the relationship between hydroxylation levels and collagen properties. Additionally, we also compared with samples hydroxylated by proline hydroxylase BaP4H.

## 2. Materials and Methods

### 2.1. Construction of Engineered Strains

Using CRISPR-Cas9 gene editing, the *proC*, *ompT*, and *lon* genes were knocked out in *E. coli* strain MG1655, resulting in an *E. coli*–proline auxotroph. Specifically, single-guide RNA (sgRNA) targeting a 20 bp recognition site within the gene to be knocked out was designed using the CRISPOR website to identify the site. And homology arm consisting of 500 bp fragments upstream and downstream of the target gene was constructed. These two fragments were introduced into the strain containing the Cas9 for gene knockout. The recombinant human collagen (rhCol), characterized by a molecular weight of 55 kDa and a proline content of approximately one-eighth of the total amino acids, was successfully cloned into the pET28a plasmid, resulting in the protein expression plasmid pET28a-rhCol, which was obtained from our research group [22]. The T7 RNA polymerase gene from *E. coli* BL21 (DE3) was cloned into the SalI/HindIII site of plasmid pBAD33 (Appendix A).

### 2.2. Expression of Collagen by Incorporating Hyp/Pro

For expression, the constructed plasmids pET28a-rhCol and pBAD33-T7RNAP were transformed into *E. coli* MG1655 △*proC*△*ompT*△*lon*. Transformed colonies were coated on Luria-Bertani (LB) agar plates containing 50 μg/mL kanamycin and 25 μg/mL chloramphenicol, and then a single colony was selected and inoculated into liquid LB medium for overnight culture at 37 °C. The seed culture was then transferred into fresh LB liquid medium and incubated for 3 to 6 h until the optical density at 600 nm (OD_600_) reached approximately 1–4. L-Arabinose was added to the medium to induce the expression of T7 RNA polymerase. After incubation for 15 mins, the culture was centrifuged to obtain the cell pellet, and then the cells were washed twice with an M9 basic medium, which contained 19 basic amino acids (not including proline among 20 basic amino acids) 100 mg/L, glucose 1%, and NaCl 500 mM. The cells were resuspended in an equal volume of the above medium and subjected to starvation to deplete intracellular residual proline in 1 h. Subsequently, isopropyl β-D-1-thiogalactopyranoside (IPTG) was added to induce the expression of rhCol. Different molar ratios of Pro and Hyp were introduced to the culture, with a total concentration of 20 mM. After an additional incubation for 3 to 6 h at 37 °C, the bacteria were collected by centrifugation for further analysis and purification of the recombinant protein.

### 2.3. Expression of Hydroxylated Collagen by Co-Expression of Proline Hydroxylase

Proline hydroxylase expression plasmid pGro7-BaP4H and plasmid pET28-rhCol, previously constructed in the laboratory, were co-transformed into *E. coli* BL21 (DE3), according to our previous report [22]. The transformed colonies were selected on LB-agar plates supplemented with 50 μg/mL kanamycin and 25 μg/mL chloramphenicol. A positive colony was chosen and inoculated into a liquid LB medium for overnight culture at 37 °C. The subsequent seed culture was transferred to fresh LB liquid medium at a 1% inoculum ratio, with the addition of 1 mg/mL L-Arabinose to induce the expression of proline hydroxylase BaP4H. The culture was cultivated at 37 °C and 220 rpm until the OD_600_ reached 0.6–0.8, at which point 1 mM IPTG was added to induce recombinant collagen expression. After 5 h of induction, the bacterial cells were collected by centrifugation.

### 2.4. Purification of Recombinant Collagen

Cells were resuspended in buffer A (20 mM Tris-HCl pH 7.4, 500 mM NaCl) and lysed by a high-pressure homogenizer. The resulting lysate underwent centrifugation at 12,000 rpm for 30 min at 4 °C, followed by filtration through a 0.22 μm membrane to obtain a clear supernatant. The supernatant was then loaded onto a 5 mL pre-packed gravity column (Ni-NTA 6FF). Recombinant collagen was purified using a gradient elution method with Buffer A supplemented with imidazole (from 50 to 500 mM). The protein was collected and the purity was analyzed by SDS-PAGE.

### 2.5. Determination of Collagen Hydroxylation Rate

A stock solution of proline was prepared at a concentration of 20 mg/mL using distilled water and subsequently diluted into 1, 2, 4, 6, 8, 10, 15, and 20 mg/L. For the colorimetric assay, 0.2 mL of each proline solution was mixed with 0.1 mL of acidic ninhydrin solution (2.5 g ninhydrin was dissolved in, 60 mL of glacial acetic acid and 40 mL of 6 mol/L phosphoric acid) and 0.1 mL of glacial acetic acid. The reaction mixture was heated in a water bath at 100 °C for 1 h and then rapidly cooled on ice to terminate the reaction. 0.4 mL of xylene was added to the reaction mixture and placed at room temperature for 20 min. After stratification, the upper solution was taken to measure the absorbance at 520 nm in an ultraviolet spectrophotometer, and the standard curve for proline quantification was drawn according to a previous study [31].

Hydroxyproline stock solution (10 mg/mL) was prepared in distilled water and diluted to 1, 2, 4, 6, 8, and 10 mg/L hydroxyproline solution. 0.2 mL hydroxyproline solution at different concentrations was mixed with 0.1 mL of chloramine T solution and incubated at room temperature for 20 min. Subsequently, 0.1 mL of a chromogenic reagent, prepared by dissolving 10 g p-dimethylaminobenzaldehyde in 35 mL perchloric acid and 65 mL isopropanol, was added to the mixture, kept in a water bath at 60 °C for 20 min, and then quickly cooled. The absorbance of the resulting solution was measured at 558 nm, and the data were plotted to generate a standard curve for hydroxyproline quantification according to a previous study [32].

Collagen samples were hydrolyzed with HCl (6 M) acid at 100 °C for 16 h, then dried on a porcelain plate to remove excess water, and the acid-free hydrolyzed sample was dissolved in 1 mL distilled water. The absorbance of a 0.2 mL sample was measured at 520 nm and 558 nm, corresponding to the specific wavelengths for Pro and Hyp detection, respectively. Subsequently, these absorbance values were applied to their respective standard curves to determine the concentrations of Pro and Hyp in the collagen samples. The hydroxylation rate of collagen was calculated by the formula (hydroxylation rates = [Hyp/(Pro + Hyp)] × 100%).

### 2.6. LC-MS/MS Analysis for Recombinant Collagen

Purified protein samples were reduced by the addition of 4 μL 0.05 M TCEP solution and incubated at 60 °C for 1 h. After reduction, 2 μL of 55 mM MMTS solution was added to the mixture and alkylated at room temperature in the dark for 45 min. The treated proteins were then digested with 50 μL of 0.5 M TEAB and 2% trypsin, incubated overnight at 37 °C. Then, an additional 1% trypsin was added to the mixture, and it was continuously incubated for another 4 h at 37 °C. After digestion, the supernatant was collected by centrifugation and vacuum-dried at a low temperature. The peptides were resuspended and centrifuged at 13,200 rpm for 20 min at 4 °C. The supernatant was collected and analyzed with a Thermo Scientific Q Exactive (Thermo Fisher Scientific, Waltham, MA, USA) mass spectrometer for online mass spectrometry.

### 2.7. CD Spectrometry Analysis

Recombinant collagen samples, prepared via various methods with distinct hydroxylation rates, were analyzed for their triple helix structure at a concentration of 1 mg/mL in 20 mM phosphate buffer using a ChirascanVX (Applied Photophysics Ltd., Leatherhead, UK) spectropolarimeter. Before analysis, the collagen solutions were dialyzed with 20 mM phosphate buffer and concentrated in a 10 kDa Spin-XR UF 500 centrifuge concentrator to a final concentration of 1 mg/mL and then stored at 4 °C. The structure of the recombinant collagen was evaluated using circular dichroism (CD) spectroscopy. This technique was applied over a wavelength span from 190 to 250 nm using a 1 mm quartz cuvette. During the process, the temperature was consistently maintained at 4 °C. The measurements were taken with a wavelength increment of 1 nm and an averaging interval of 1 s for each step, and the results were derived from the mean of three consecutive scans.

The recombinant collagen with different preparation methods was subjected to thermal denaturation analysis. The recombinant collagen sample, placed in the quartz cuvette, underwent a temperature increase from 4 °C to 50 °C at a rate of 1 °C per min. Throughout this heating process, the circular dichroism (CD) signal intensity at a wavelength of 221 nm was continuously monitored and recorded.

### 2.8. Scanning Electron Microscopy (SEM) Analysis

After drying, the purified recombinant collagen, exhibiting varying degrees of hydroxylation, underwent dialysis using a 10 mM phosphate buffer (PB) solution. Following dialysis, the samples were applied onto the silicon wafer and subjected to freeze-drying to remove moisture. After freeze-drying, the samples were coated with a thin layer of gold for 2 min to enhance conductivity and visual clarity. The morphology of the samples was then scrutinized using scanning electron microscopy.

### 2.9. Cell Binding Assay

Mouse fibroblast cells (NIH/3T3) were utilized to evaluate the cell binding of different recombinant collagen. After dialysis in phosphate-buffered saline (PBS), the collagen samples were adjusted to a concentration of 0.5 mg/mL. Next, 30 μL of each collagen solution were aliquoted into individual wells of a 96-well plate. Subsequently, the plate was incubated for an hour within a cell culture setting. After incubation, 100 μL of NIH/3T3 cell suspension, having a density of 1 × 10^5^ cells/mL under DMEM medium, were added to each well. Then, the plate was placed in an incubator at 37 °C with 5% CO_2_ for 60 min to promote cell adhesion. The medium was carefully aspirated, and each well was washed three times with PBS to remove non-adherent cells, and the residual liquid was sucked out as much as possible. The cell binding was quantified by measuring the absorbance at 450 nm using a CCK-8 kit (Biosharp, Beijing, China).

### 2.10. Cell Viability Assay

Mouse fibroblast cells (NIH/3T3), when they reached a confluence of 70% to 80% in their culture dish, were treated with trypsin to a suspension with a cell density of 5 × 10^4^ cells/mL. 100 μL of the cell suspension was inoculated into each well in a 96-well plate and incubated for 12 h under standard culture conditions (37 °C, 5% CO_2_, saturated humidity). After the removal of the culture medium, 100 μL of recombinant collagen diluted to a final concentration of 0.1 mg/mL using DMEM medium was added into each well and incubated in an incubator at 37 °C, 5% CO_2_ saturated humidity for 48 h. Subsequently, cell viability was assessed using a CCK-8 kit with absorbance measured at 450 nm.

### 2.11. Fermentation of Collagen by Incorporation in a 7 L Fermenter

The seed culture was inoculated into a 7 L fermenter containing 1800 mL of fermentation medium, which consisted of glycerol 30 g/L, yeast extract 50 g/L, (NH_4_)_2_SO_4_ 6.12 g/L, KH_2_PO_4_ 5.85 g/L, MgSO_4_(7H_2_O) 1 g/L, and EDTA 1 g/L. The inoculation was performed at a rate of 10% of the total medium volume. During fermentation, the temperature was kept at 37 °C, and the aeration was adjusted to a flow rate of 4 m^3^/(min × m^3^) to facilitate efficient oxygen transfer. The agitation speed was dynamically controlled to sustain the dissolved oxygen (DO) level around 30%. Additionally, the pH of the fermentation broth was stabilized at around 6.8 through the automated titration of ammonia solution and hydrochloric acid. To monitor the fermentation, samples of fermentation broth were taken and subjected to analysis at one-hour intervals. Upon approaching the stationary phase of cell growth, L-Arabinose was added into the broth to induce the expression of T7 RNA polymerase. After induction for 1 h, the bacterial cells were harvested by centrifugation under sterile conditions, yielding a pellet. This pellet was then resuspended in an equal volume of sterile M9 fermentation medium, which was composed of the M9 basic medium along with 18 basic amino acids (not including proline and glycine among 20 basic amino acids) at a concentration of 400 mg/L, glycine at 2 g/L, glucose at 1%, and NaCl at 500 mM. The resuspended cells were reintroduced into the 7 L fermenter, and the fermentation process was continued under the same conditions as before. After starvation for 1 h, the mixture, including 1 mM IPTG, 16 mM Pro, and 24 mM Hyp, was added to the culture. Subsequently, a 50 mL aliquot of feed medium, which contained 4 mM of Pro, 6 mM of Hyp, and a mixture of 18 essential amino acids (not including proline and glycine among 20 basic amino acids) at a concentration of 2 g/L, along with 10 g/L glycine, was supplemented into the fermenter at hourly intervals.

## 3. Results and Discussion

### 3.1. Construction of Engineering Bacteria

Firstly, a proline nutrient-deficient strain *E. coli* MG1655 △*proC*△*ompT*△*lon* was constructed, and a co-expression system involving T7 RNA polymerase and pET series plasmids was also established. Through CRISPR-Cas9 gene editing technology, the gene *proC* was targeted and knocked out in the *E. coli* MG1655 strain, a gene that encodes a crucial enzyme involved in the biosynthesis of proline (as depicted in Figure 2A). To confirm the proline auxotrophy of the *E. coli* MG1655 △*proC*, the strain was cultured in an M9 basic medium on a shaking incubator. The engineered strain cannot grow without the addition of exogenous proline, demonstrating the successful development of a proline-deficient strain (illustrated in Figure 2B).

Following the initial genetic modification, steps were taken to further boost recombinant protein production in the engineered *E. coli* strain. ompT and lon were systematically inactivated; specifically, the protease-encoding genes *ompT* and *lon* were knocked out systematically within the genome of the previously developed *E. coli* MG1655 △*proC* (as shown in Figure 2C). Concurrently, a co-expression system consisting of T7 RNA polymerase and a T7 promoter was introduced; this integration enhanced the expression levels of the recombinant protein by providing a more efficient transcriptional machinery. The gene encoding recombinant collagen rhCol was inserted into the plasmid pET28a. Concurrently, the gene of T7 RNA polymerase was housed within the plasmid pBAD33. Both plasmids were then introduced into *E. coli* MG1655 △*proC*△*ompT*△*lon,* thereby creating the engineered bacterial strain. The bacterial seed solution was cultivated in an LB medium, while the protein expression was induced by L-Arabinose and IPTG. Subsequent analysis by SDS-PAGE and gray-scale scanning indicated that the yield of rhCol was approximately 50% higher after the systematic knockout of the two protease genes. The observed enhancement in protein yield elevated the levels to be comparable to those produced by the benchmark *E. coli* BL21 (DE3) strain. This achievement substantiated the successful optimization of our engineered strain, which was specifically tailored for enhanced expression of recombinant proteins, as evidenced in Figure 2D,E.

### 3.2. Incorporations of Pro and Hyp

Previous studies have demonstrated that the addition of exogenous Hyp facilitates the co-translation of Hyp with intracellular Pro during cell culture, and the concentrations of incorporated Hyp and NaCl have a significant impact on the efficiency of protein expression [29]. In this study, a novel system was developed for the preparation of recombinant collagen with varying degrees of hydroxylation. This was achieved by adjusting the ratios of Pro to Hyp in the culture medium during the synthesis process, as illustrated in Figure 3A. In the absence of proline, the M9 basic medium was supplemented with varying concentrations of NaCl and Hyp to investigate their effects on the expression of rhCol in this incorporation system. It was observed that, with a fixed NaCl concentration of 500 mM, the expression of rhCol increased as Hyp concentrations increased, achieving the highest value when Hyp was at 20 mM, as depicted in Figure 3B. Furthermore, when the concentration of Hyp was fixed at 20 mM, the expression of rhCol increased and then declined as NaCl concentration was progressively elevated, peaking at a NaCl concentration of 500 mM, as shown in Figure 3C. This observed pattern might be correlated with the capacity of NaCl to increase the osmotic pressure in the culture medium, which in turn could enhance the uptake of Hyp and the subsequent synthesis of collagen.

Several parameters, including the biomass before induction, culture time after induction, the concentrations of Pro and Hyp, and the ratio of Pro to Hyp, were explored. As shown in Figure 3D,E, it showed that the OD_600_ in LB medium before centrifugation and the incubation time after induction in M9 had a small effect on the hydroxylation rate. However, these two factors greatly affect the protein expression. Thus, the OD_600_ before centrifugation can be set at 4, and the incubation time after induction can be maintained at 6 h in a shake flask. When proline hydroxylase was co-expressed with recombinant collagen in a previous report, the hydroxylation rate fluctuated over time with a maximum difference of up to 20% [22]. The activity of this enzyme is not stable when there is a change in the conditions of the culture medium, leading to a substantial impact on the rate at which recombinant collagen undergoes hydroxylation. The hydroxylation rate of recombinant collagen increased as the concentration of Hyp rose and decreased as the concentration of Pro rose. The hydroxylation rate of rhCol was in the range of 40–50% when the concentrations of Pro and Hyp were set at 12 mM and 8 mM, respectively, as depicted in Figure 3F. After purification via a one-step Ni-affinity column, unhydroxylated and four rhCols with incremental hydroxylation rates of approximately 20% (24%, 43%, 64%, 88%) were obtained. Furthermore, hydroxylated recombinant rhCol was also successfully produced by co-expressing it with proline hydroxylase BaP4H, the resulting hydroxylation rate of rhCol by this method was found to be 63%.

### 3.3. The Structural Characteristics of rhCol

During the preparation of hydroxylated recombinant collagen, the incorporation method leads to the random substitution of Pro by Hyp during the translation stage of collagen peptides [29]. However, proline hydroxylase P4H selectively catalyzes the hydroxylation of Pro residues in the Y position of the Gly-X-Y motif in collagen according to the characteristics of this enzyme. To validate this hypothesis, hydroxylated rhCol prepared by two different methods were analyzed for hydroxylation sites using LC-MS/MS (Figure 4A). The rhCol produced by the incorporation displayed a random distribution of Hyp across the Pro sites throughout the entire collagen peptide sequence, with each site having an equal chance of being substituted. In contrast, the rhCol hydroxylated by the enzyme BaP4H exhibited a specific pattern, with Hyp being incorporated exclusively at the Y position within the Glycine-X-Y sequence of the collagen structure. It was found that Pro at the Y position within the Gly-X-Y sequence of the rhCol sequence constituted about two-thirds of the total Pro in the sequence. Furthermore, the 3-Hyp at the X position plays a crucial role in the lateral association of collagen molecules and is essential for the initial assembly of collagen fibrils with a D stagger of molecules [33]. The outward orientation of 3-Hyp within the triple helix could enable the formation of short-range hydrogen bonds between individual triple helices. This interaction is likely to enhance the supramolecular assembly and may significantly contribute to the overall stability of the triple helix conformation.

Subsequently, rhCol with varying hydroxylation levels prepared by incorporation of Hyp/Pro and rhCol hydroxylated by BaP4H were detected by circular dichroism (CD) to examine their triple-helical conformation and thermal stability. The CD assay conducted at 4 °C revealed that hydroxylated rhCol exhibited a significantly pronounced positive absorption peak at 221 nm, as depicted in Figure 4B. In contrast, the unhydroxylated rhCol demonstrated a CD value that was nearly 0 mdeg. For rhCol with a 24% hydroxylation rate, the CD value was around 2.5 mdeg, and this value increased to 7.5 mdeg for rhCol with a 43% hydroxylation rate. It is noteworthy that further increases in hydroxylation rate did not lead to significant increases in CD values. Although rhCol hydroxylated by the enzyme BaP4H reached a 63% hydroxylation rate, it only exhibited a CD value of about 3.5 mdeg. It is acknowledged that a standard collagen CD spectrum should feature a negative peak around 195 nm and a positive peak around 221 nm. The absence of a positive peak at this wavelength is indicative of an absent triple-helical structure [34,35,36,37]. Therefore, it is deduced that rhCol possesses the structural characteristics of a triple helix after hydroxylation. Notably, the rhCol prepared through incorporation, even with a lower degree of hydroxylation at 43% compared to collagen hydroxylated by BaP4H, demonstrated more pronounced characteristics indicative of a triple-helical conformation. This discrepancy could be due to the random substitution of Pro at both X and Y positions in the Gly-X-Y sequence with Hyp, in contrast to the selective hydroxylation of Pro at the Y position by the enzyme BaP4H. This non-selective replacement may enhance the possibility of forming a stable triple-helix structure.

Additionally, to evaluate the thermal stability of the recombinant collagen variants, a CD thermal melting assay was conducted. The thermal denaturation of collagen is characterized by irreversible and time-dependent transitions, which transform the molecule from its native triple-helical conformation to a disordered state [38]. Thermal melting analysis indicated that non-hydroxylated rhCol had a melting temperature (Tm) of 22 °C, while the Tm for rhCol hydroxylated by the enzyme BaP4H was 25 °C. Both results were lower than a Tm of 27 °C observed for rhCol prepared by incorporation, which had achieved a 43% hydroxylation rate, as illustrated in Figure 4C. The CD intensity of these collagens experiences significant variation with temperature increases at lower temperatures, and the rate of change slows down as the temperature approaches the Tm. It is noteworthy that rhCol prepared by incorporation showed a higher CD value at the Tm, suggesting a greater likelihood for refolding into its native structure upon cooling. The thermal stability of collagen is mainly determined by interchain hydrogen bonds within the triple-helical framework. The temperature at which collagen begins to denature, known as the melting temperature (Tm), depends on the nucleation phase during collagen fibril formation. The presence of Hyp is recognized to reinforce these hydrogen bonds, which in turn strengthen the triple-helical structure and improve the collagen’s resistance to heat [39,40].

Subsequently, three rhCols with different hydroxylation rates by incorporation method were analyzed by SEM. SEM scans revealed that all samples are self-assembled into network structures, with the hydroxylated rhCol exhibiting a more uniform and smooth fibrous structure. It is particularly noteworthy that when compared to hydroxylated rhCol depicted in Figure 4D, rhCol, with a 43% hydroxylation rate, as shown in Figure 4E, displayed smaller pore sizes and smoother fiber. When the degree of hydroxylation surpassed a certain threshold, rhCol exhibited irregular pore sizes and formed a sheet-like network structure (as illustrated in Figure 4F). Within the triple helix configuration, the hydroxyl group of hydroxyproline extends outward, serving the dual role of a hydrogen bond donor and acceptor. This capacity is instrumental in forming a water bridge, which significantly contributes to the interaction between other triple helix molecules, as cited in studies [41,42,43]. The increase in hydrodynamic radius strongly indicates that the presence of Hyp in rhCol could enhance intermolecular interaction between each domain, thereby facilitating the formation of fibers [44]. This observation suggests that the optimal degree of hydroxylation is crucial for promoting intermolecular interactions of collagen and the formation of fibrous structures. When the hydroxylation rate of rhCol reaches 43%, the intermolecular interactions may be maintained at an optimal level, thereby promoting the formation of its fibrous structure and enhancing the stability of its conformation. However, as the hydroxylation rate of rhCol gradually increases, the most stable intermolecular forces are disrupted, thereby affecting the structural stability and forming a structure similar to that of unhydroxylated rhCol.

### 3.4. Cell Binding and Viability of rhCol

Here, the impacts of different hydroxylation rates of rhCols, which were prepared by incorporating Hyp/Pro from the culture medium, on the adhesion and viability of mouse fibroblasts NIH/3T3 were investigated, as well as the rhCol with optimal hydroxylation rate was compared with that of rhCol hydroxylated by hydroxylase and the standard bovine type I collagen. Initially, a range of rhCol samples with different hydroxylation levels, produced through the incorporation technique, underwent cell binding assays, with the unhydroxylated rhCol (0% hydroxylation) serving as a control (as shown in Figure 5A). A significant improvement in cell binding was observed after recombinant collagen was hydroxylated. It is intriguing to note that for rhCol, a moderate level of hydroxylation was more conducive to cell attachment within a certain range rather than the absolute maximum. The rhCol sample with the highest cell-binding affinity had a hydroxylation rate of 43%, which is close to the natural hydroxylation rate of human collagen, and it showed nearly a 30% increase in cell binding compared to unhydroxylated rhCol. Subsequently, the rhCol with 43% hydroxylation was selected for a comparative study of cell binding with rhCol hydroxylated by the enzyme BaP4H and with standard bovine type I collagen. For this analysis, the relative cell binding of the latter two types of collagens was calculated relative to the rhCol with 43% hydroxylation, which served as the reference control (as depicted in Figure 5B). Both hydroxylated rhCols showed an increase in cell binding when compared to the non-hydroxylated one. However, the rhCol sample with a 43% hydroxylation rate, which was prepared through the incorporation method, exhibited a significantly higher enhancement in cell binding. This improvement was so pronounced that it even exceeded that of the standard bovine type I collagen. As for the effect of hydroxylated collagen on biological activity, the coevolution of vertebrate collagens and the αI domain containing integrins has led to a special subgroup of receptors that has the ability to recognize GXXGER type triple-helical motifs. The high avidity-binding site for DDR-type collagen receptors contains Hyp too. These receptors may mediate cell adhesion to fully maturated collagen fibrils in tissues [45].

In addition, the effect of these proteins on cell viability was also investigated. Similarly, the relative cell viability of collagen samples with different hydroxylation rates was calculated using unhydroxylated rhCol as a control (as shown in Figure 5C). Contrary to expectations, the hydroxylation of rhCol did not stimulate cell proliferation. Instead, it was found to impede cell growth once the hydroxylation rate exceeded a certain point. This inhibitory effect might be attributed to the formation of a collagen structure with excessive hydroxyproline, which could confer toxicity and be harmful to the growth of NIH/3T3 cells. The above results contradict previous studies that reported a higher degree of recombinant collagen, achieved by incorporating large amounts of Hyp, could improve cell viability [29]. It indicates that the influence of hydroxylation on the characteristics of recombinant collagen has a complicated relationship with its own structure. Then, the rhCol with a 43% hydroxylation rate was also chosen for a comparative analysis of cell viability with rhCol hydroxylated by the enzyme BaP4H and standard bovine type I collagen, as depicted in Figure 5D. The cell viability of rhCol hydroxylated by the two hydroxylation methods was found to be comparable, but it was noted to be a little lower than that of the standard bovine type I collagen.

### 3.5. Enhanced Production of rhCol in the 7 L Fermenter

The technique for producing hydroxylated recombinant collagen by exogenous incorporation has been successfully scaled up to a 7 L fermenter. The process was divided into two stages: the first stage was dedicated to the proliferation of a substantial cell mass, and the subsequent phase focused on obtaining enhanced production of recombinant collagen through the supplementation of amino acids. The ratio of Pro to Hyp in the culture during the fermentation was 4:6. Fermentation in a 7 L fermenter demonstrated that a rapid increase in the expression of hydroxylated recombinant collagen was found after induction with IPTG, which subsequently plateaued, culminating in a final protein yield of 1.186 g/L, as illustrated in Figure 6A,B. After cell lysis and protein purification were completed, the hydroxylation rate of the final product was determined to be 55%, which is comparable to the results from fermentation in a shake flask. It indicates that the exogenous incorporation method is just as equally efficient for the production of hydroxylated recombinant collagen in a fermentation tank. However, the two-stage fermentation process presents challenges, including operational complexity and a higher risk of contamination. To achieve a more efficient, cost-effective, and manageable production process, it is essential to further optimize the fermentation protocol.

## 4. Conclusions

In this study, an engineered strain deficient in proline was constructed for efficient production of recombinant collagen. Concurrently, a system was established for the preparation of hydroxylated recombinant collagen through the incorporation of exogenous Pro and Hyp. This approach enables the preparation of recombinant collagen with different hydroxylation rates by adjusting the concentrations of Pro and Hyp in the culture medium. Utilizing this technique, several recombinant human collagens (rhCol) with different hydroxylation rates were produced. Subsequently, a comparative study was conducted to examine the correlation between their hydroxylation levels, structural properties, and biological performance. A hydroxylation rate of 43% was determined to be ideal, closely approximating the level of native human collagen. rhCol that was hydroxylated through incorporation showed a 29% enhancement in cell binding compared to unhydroxylated rhCol. Additionally, the melting temperature (Tm) of this hydroxylated rhCol was elevated by 5 °C, and all proline in the collagen sequence could be converted to hydroxyproline. However, cell viability analysis revealed that an excessively high rate of hydroxylation unexpectedly hinders cell proliferation. This finding could be interpreted from an evolutionary perspective to rationalize the natural collagen hydroxylation rate in humans, which is maintained at a moderate level of 40–50%, avoiding both higher or lower. The system achieved 1.186 g/L of rhCol in a batch-fed manner in a 7 L fermenter. Further research is needed to explore the relationship between collagen hydroxylation, its structural characteristics, and functional properties. This will help to elucidate the fundamental mechanisms and enhance the manufacturing process of recombinant collagen, which is crucial for its use in the biomedical field.

## Figures and Tables

**Figure 1 bioengineering-11-00975-f001:**
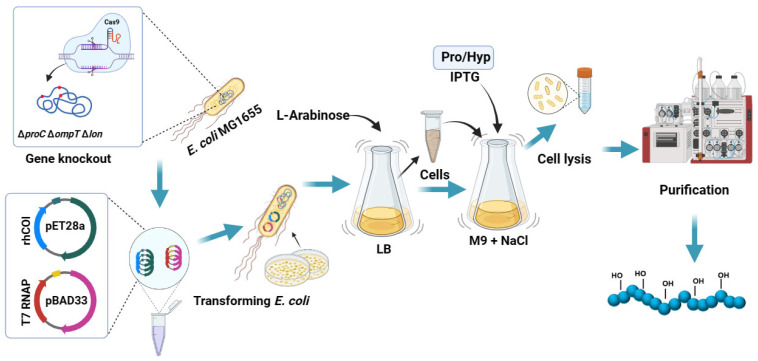
Schematic diagram of preparation for hydroxylated recombinant collagen by incorporating Pro and Hyp.

**Figure 2 bioengineering-11-00975-f002:**
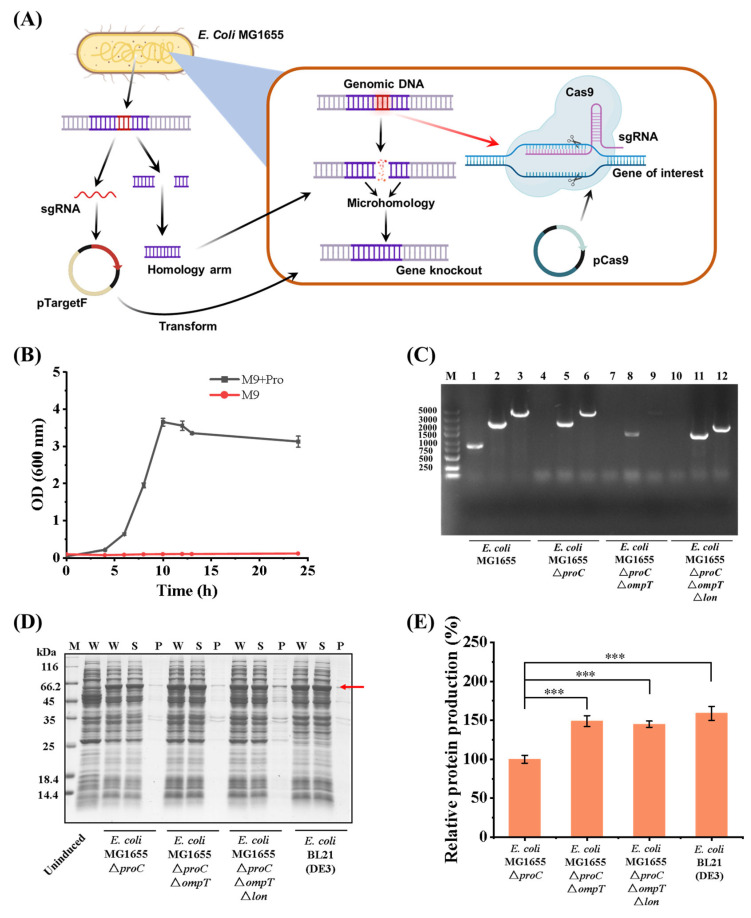
Construction and verification of engineering bacteria. (**A**) Schematic of CRISPR-Cas9 gene editing technology for gene knockout. (**B**) Growth curve of *E. coli* MG1655 △*proC* in M9 basic medium. The black line refers to the addition of 120 mg/L of proline to the M9 basic medium. (**C**) Agarose gel electrophoresis analysis for gene knockout verification of proline-deficient strains. M: marker, Lanes 1, 4, 7, 10: verification of *proC* gene knockout; Lanes 2, 5, 8, 11: verification of *ompT* gene knockout; Lanes 3, 6, 9, 12: verification of *lon* gene knockout. (**D**) SDS-PAGE analysis of protein rhCol expression in different strains. M: protein marker, W: protein in whole cell solution, S: protein in supernatant, P: protein in precipitation. The red arrow indicates the location of the target protein. (**E**) Relative expression of the protein rhCol in different strains. (Each test with three replicates (n = 3). Data were analyzed by a one-way ANOVA using SPSS (version 22), and the results are presented as mean ± SD, *** *p* < 0.001).

**Figure 3 bioengineering-11-00975-f003:**
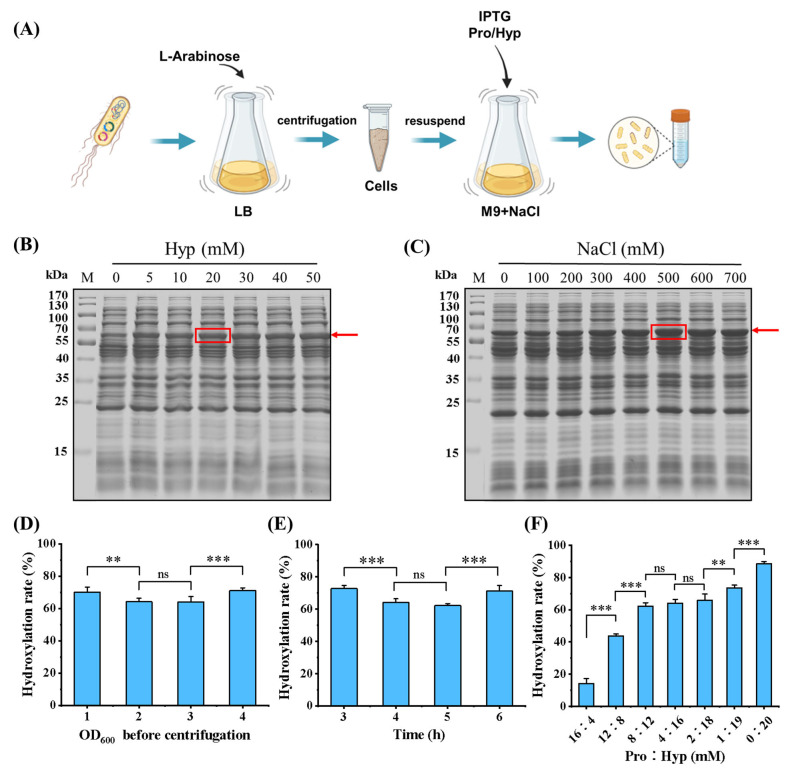
Preparation of hydroxylated recombinant collagen by incorporation of Pro and Hyp from culture medium. (**A**) Flow chart of the novel system for preparation of hydroxylated recombinant collagen. (**B**) SDS-PAGE analysis of protein expression with different concentrations of Hyp incorporated. M: protein marker. The red arrow indicates the location of the target protein, and the red box highlights the most appropriate Hyp concentration. (**C**) SDS-PAGE analysis of protein expression with different concentrations of NaCl in M9 basic medium. M: protein marker. The red arrow indicates the location of the target protein, and the red box highlights the most appropriate Nacl concentration. (**D**) The effect of different OD_600_ in LB medium before centrifugation on hydroxylation of recombinant collagen. In the culture medium, the concentrations of Pro and Hyp were 4 mM and 16 mM, respectively. The culture time after induction was 4 h. (**E**) The effect of different culture times after induction in M9 on hydroxylation of recombinant collagen. In the culture medium, the concentrations of Pro and Hyp were 4 mM and 16 mM, respectively. The OD_600_ before centrifugation was 3. (**F**) The effect of different ratios of Pro to Hyp on hydroxylation of recombinant collagen (each test was performed with three replicates (n = 3). Data were analyzed by a one-way ANOVA using SPSS, and the results are presented as mean ± SD, ns > 0.05, 0.001 < ** *p* < 0.01, *** *p* < 0.001).

**Figure 4 bioengineering-11-00975-f004:**
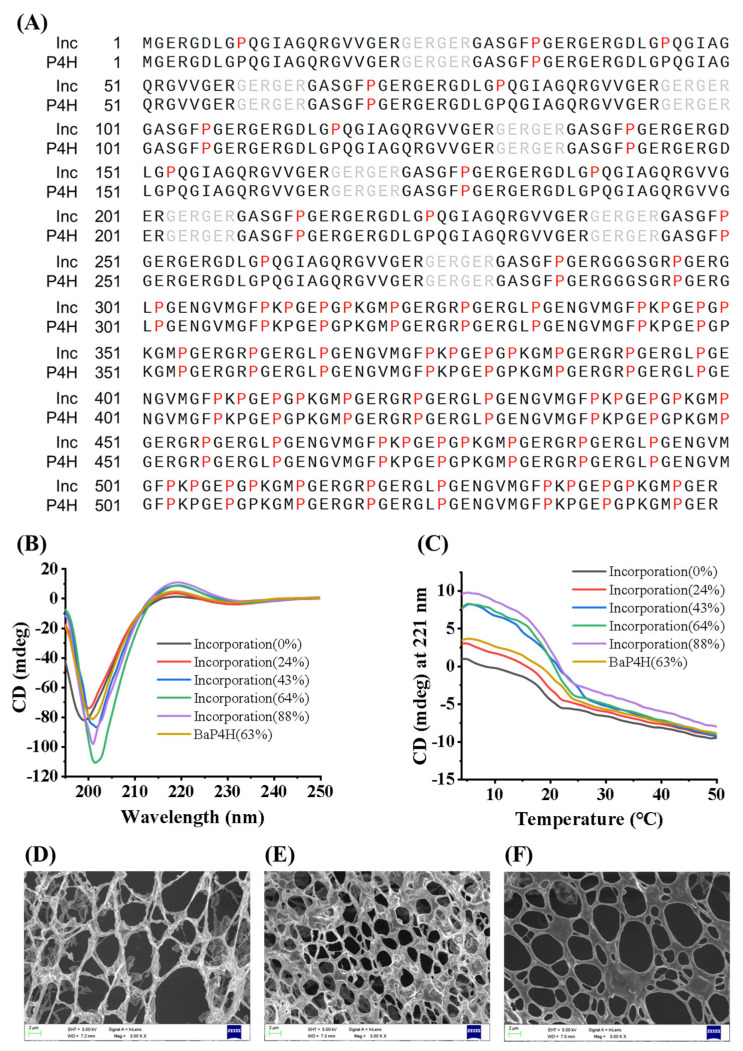
Structural analysis of rhCol. (**A**) Identification of hydroxylation sites in rhCol by LCMS/MS. “P” marked in red refers to hydroxylated proline; the gray part refers to the sequence that failed to be detected; “Inc” refers to rhCol prepared by incorporation; “P4H” refers to rhCol hydroxylated by proline hydroxylase BaP4H. (**B**) CD spectra of rhCol with different hydroxylation rates and different hydroxylation methods. The hydroxylation resulted in an obvious triple helix positive absorption peak at 221 nm. (**C**) CD thermal melting analysis of different rhCols. (**D**) SEM analysis of unhydroxylated rhCol. (**E**) SEM analysis of rhCol with a 43% hydroxylation rate. (**F**) SEM analysis of rhCol with an 88% hydroxylation rate.

**Figure 5 bioengineering-11-00975-f005:**
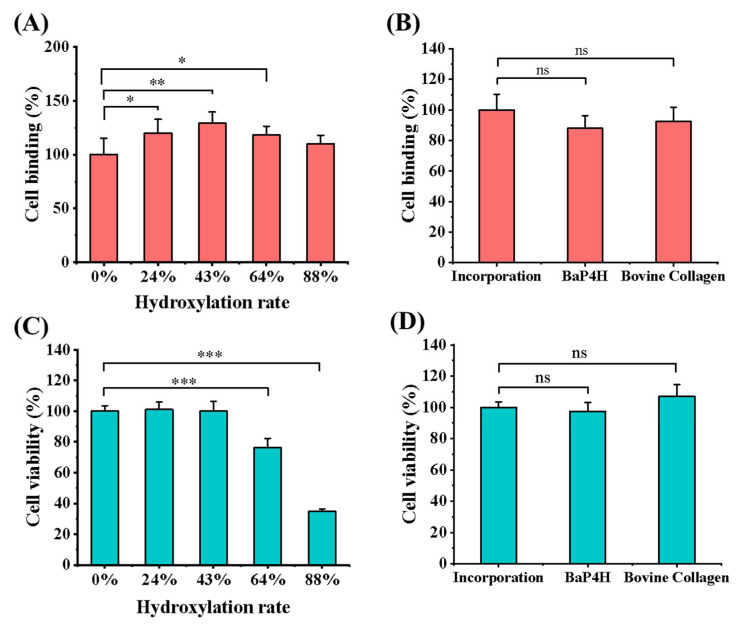
Biological characterization of different rhCol samples. (**A**) Cell binding analysis of rhCol with different hydroxylation rates. These rhCol samples were prepared through the incorporation method. (**B**) Cell binding analysis of three collagens with nearly the same hydroxylation rate of around 43%, prepared through the incorporation method, rhCol hydroxylated by BaP4H, standard bovine type I collagen. (**C**) Cell viability analysis of rhCol with different hydroxylation rates. These rhCol samples were prepared through the incorporation method. (**D**) Cell viability analysis of three collagens with nearly the same hydroxylation rate of around 43%, prepared by incorporation method, rhCol hydroxylated by BaP4H, standard bovine type I collagen. Each test with three replicates (n = 3). Data were analyzed by a one-way ANOVA using SPSS, and the results are presented as mean ± SD, ns > 0.05, 0.01 < * *p* < 0.05, 0.001 < ** *p* < 0.01, *** *p* < 0.001.

**Figure 6 bioengineering-11-00975-f006:**
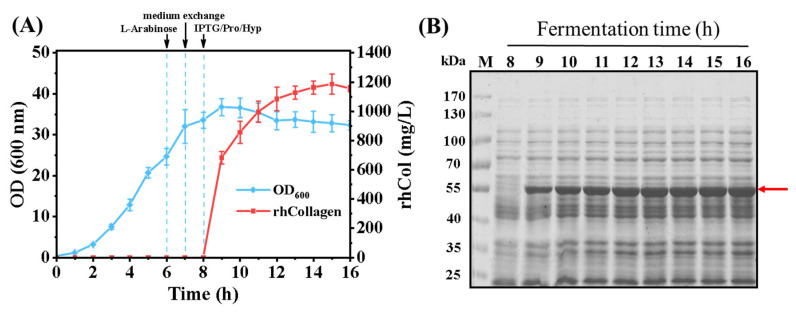
The fed-batch fermentation of hydroxylated recombinant collagen by incorporation in a 7 L fermenter. (**A**) Cell growth curve (blue) and production of hydroxylated rhCol (red) in a 7 L fermenter. Arrows represent the transition point for protein induction and cultivation stage, L-Arabinose: the time of inducing T7RNA polymerase expression; medium exchange: the point of replacing fermentation medium with M9 fermentation medium; IPTG/Pro/Hyp: the time of inducing rhCol expression. (**B**) SDS-PAGE analysis of the production of hydroxylated rhCol during fermentation in a 7 L fermenter. M: protein marker.

## Data Availability

Data will be made available on request.

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
