# Peer review of "Preparation and Characterization of Hydroxylated Recombinant Collagen by Incorporating Proline and Hydroxyproline in Proline-Deficient Escherichia coli"

_bioengineering, 2024, doi:10.3390/bioengineering11100975_

Round 1

Reviewer 1 Report

Comments and Suggestions for Authors

A nice work relating to characterization of hydroxylated recombinant collagen has been presented. Suitable characterizations and results can be found in the manuscript. However, there are some points which should be considered in the revised version for further completion of the work. Therefore, I suggest revision of the manuscript based on addressing to the following points:

1.       The first sentence of the introduction section can be further supported by, e.g., Prevascularized micro-/nano-sized spheroid/bead aggregates for vascular tissue engineering, for further completion of the literature review regarding the application of collagen in tissue engineering.

2.       E. coli bacteria can be also used for growth and mutation of viruses (such as bacteriophage viruses), as a suitable natural culture media. See, for example, Protein degradation and RNA efflux of viruses photocatalyzed by graphene-tungsten oxide composite under visible light irradiation. This point should be mentioned in the revised version.

3.       In addition to the P-values, the number of biological tests should be given in the caption of the related figures.

4.       Computational methods for designing the recombinant proteins applicable in anticancer (Anticancer drug discovery based on natural products: from computational approaches to clinical studies), antiviral (A review on computer-aided chemogenomics and drug repositioning for rational COVID-19 drug discovery) and recently anti CRISPR (Rethinking protein drug design with highly accurate structure prediction of anti-CRISPR proteins) have been attracted the attention of the researchers in the field of drug discovery. These should be addressed and discussed in the introduction section of the manuscript.

Comments on the Quality of English Language

Minor editing of English language required.

Author Response

Comments 1: The first sentence of the introduction section can be further supported by, e.g., Prevascularized micro-/nano-sized spheroid/bead aggregates for vascular tissue engineering, for further completion of the literature review regarding the application of collagen in tissue engineering.

Response 1: The first sentence intends to state the importance of collagen, the cited literatures are reviews. The recommended reference was about the application of nano-micro size aggregations on the prevascularizations and biomedicine, very impressive and innovative work. But it does not have a strong relevance to the application or expression of collagen, hence it is inappropriate to cite this paper here.

Comments 2: E. coli bacteria can be also used for growth and mutation of viruses (such as bacteriophage viruses), as a suitable natural culture media. See, for example, Protein degradation and RNA efflux of viruses photocatalyzed by graphene-tungsten oxide composite under visible light irradiation. This point should be mentioned in the revised version.

Response 2: E. coli indeed has many uses, including the application presented in the recommended reference. But in this article, we emphasize some advantages of E. coli as a system for the expression of heterologous proteins. For other applications of E. coli, to ensure the focus of the topic, they will no longer be mentioned.

Comments 3: In addition to the P-values, the number of biological tests should be given in the caption of the related figures.

Response 3: Thanks for your valuable suggestion. The number of biological experiments has been given in the title of the related figures. Please see “Each test with three replicates (n=3)” in the Figure 2 (line 267), Figure 3 (line 315) and Figure 5 (line 461) in manuscript.

Comments 4: Computational methods for designing the recombinant proteins applicable in anticancer (Anticancer drug discovery based on natural products: from computational approaches to clinical studies), antiviral (A review on computer-aided chemogenomics and drug repositioning for rational COVID-19 drug discovery) and recently anti CRISPR (Rethinking protein drug design with highly accurate structure prediction of anti-CRISPR proteins) have been attracted the attention of the researchers in the field of drug discovery. These should be addressed and discussed in the introduction section of the manuscript.

Response 4: We are grateful for the time you have taken to review our manuscript. The paper aims to establish an innovative system and method for expressing human collagen, which does not involve methods of protein design, nor does it involve drug discovery. We have carefully considered its relevance with the recommended references. While the work is undoubtedly significant, it does not directly contribute to the specific discussion we present in our paper. Our aim is to keep the references tightly focused on expression of recombinant collagen.

Reviewer 2 Report

Comments and Suggestions for Authors

The research article titled, "Preparation and characterization of hydroxylated recombinant collagen by incorporating proline and hydroxyproline in proline-deficient Escherichia coli", describes a recombinant process through which hydroxylated collagen can be obtained. Further, the authors have demonstrated a control over the degree of the hydroxylation, thus making this process an interesting one. The manuscript has been well written and the results support the claims. The manuscript can be accepted for publication.

Author Response

Response: Thank you very much for your positive comments about our previous submitted paper-" Preparation and characterization of hydroxylated recombinant collagen by incorporating proline and hydroxyproline in pro-line-deficient Escherichia coli" in the bioengineering-3183892.

Reviewer 3 Report

Comments and Suggestions for Authors

Dear Authors,

The work performed in this study is well-planned and in detail, however some sections need to be revised meticulously .  Please find my comments in the attached file herewith.

Author Response

Comments 1: Regarding the fermentation studies at fermenter scale, have you optimized the agitation speed, DO, growth parameters etc. As I can see only the specific conditions used for fermentation at larger scale.

Response 1: Those parameters have been optimized. Limited to the length of the paper, it is not shown in this manuscript. The specific parameters are as follows: Temperature, 37°C; Aeration rate, 4 m³/(min·m³); Agitation speed, dynamically controlled; Dissolved oxygen (DO) level, maintained around 30%; pH approximate 6.8 controlled by automated titration with ammonia solution and hydrochloric acid) that have been determined in the fermentation section.

Comments 2: As per the SEM analysis results, as shown in Fig.4D, E and F unhydroxylated (D) and hydroxylation with 88% rate (F) seems to be more or less similar morphologically. On the other hand, in case of Fig.4E with hydroxylation of 43% rate has shown a significant difference in morphology. Please explain the observation with respect to the hydroxylation rate.

Response 2: From a morphological perspective, the most significant differences lie in the pore size and distribution of collagen fibers, as shown in Fig.4D, E and F. It was demonstrated that the pore size and distribution of collagen fibers can significantly affect the mechanical properties of collagen fibers. And the pH, ionic strength and collagen structure can affect the formation of fibers (Pawelec K.M et al. Collagen: a network for regenerative medicine. J. Mater. Chem. B, 2016, 4: 6484-6496.). When the hydroxylation rate of rhCol reaches 43%, the intermolecular interactions may be maintained at an optimal level, thereby promoting the formation of its fibrous structure and enhancing the stability of its conformation. However, as the hydroxylation rate of rhCol gradually increases, the most stable intermolecular forces are disrupted, thereby affecting the structural stability and formed a structure similar to that of unhydroxylated rhCol. Please see lines 419-424 in manuscript.

Comments 3: As shown in Fig.5, the cell binding started decreasing with further increasing hydroxylation rate (after 43% hydroxylation rate). Also, cell viability starts decreasing with the same pattern. Please explain the phenomenon and discuss the correlation among the factors.

Response 3: As for the effect of hydroxylated collagen on biological activity, the coevolution of vertebrate collagens and the αI domain containing integrins has led to a special subgroup of receptors that has the ability to recognize GXXGER type triple-helical motifs. The high avidity-binding site for DDR type collagen receptors contains Hyp too. These receptors may mediate cell adhesion to fully maturated collagen fibrils in tissues (Rappu P et al. Role of prolyl hydroxylation in the molecular

interactions of collagens. Essays Biochem, 2019, 63, 325 - 335). Both cell binding and cell viability gradually decreased once the hydroxylation rate of rhCol exceeded 43%, which is close to the hydroxylation rate of natural human collagen. The possible reason is that excessive hydroxylation may destroy the structure of rhCol, thereby affecting the affinity of collagen receptors to it. For deeply understanding the relationship between hydroxylation rate and structure, further structural research is required. The cell viability results contradict previous studies (Ilamaran M et al. A self-assembly and higher order structure forming triple helical protein as a novel biomaterial for cell proliferation. Biomater Sci-UK, 2019, 7(5): 2191-2199), which reported that a higher degree of recombination in collagen, achieved by incorporating large amounts of hydroxyproline, could improve cell viability. It is highly likely due to the differences of the collagen sequence.

Comments 4: As seen in Fig. 6, IPTG was added after 8 hrs of fermentation as an inducer and initially fermentation was started with arabinose as carbon source. What is the logic behind adding IPTG intermittently instead of adding at the start of fermentation.

Response 4: From the diagram in Figure 1, it can be observed that an in vitro expression system for T7 RNA polymerase and the T7 promoter has been constructed. In this system, the expression of T7 RNA polymerase is induced firstly by L-Arabinose, and then the expression of recombinant collagen is induced by IPTG after 8 hrs. T7 RNA polymerase is an efficient enzyme derived from T7 phage. It can recognize and bind to specific T7 promoter sequences to initiate RNA synthesis.

Round 2

Reviewer 1 Report

Comments and Suggestions for Authors

The authors submitted a revised version. But, nearly no significant improvement can be found in it. Some of the points which can be listed at this stage are as follows:

Concerning Comment #1, the response of the authors is not acceptable. The mentioned reference is a review containing the important points regarding the collagen. Therefore, It should be addressed if they are discussing on the ECMs.

Concerning Comment #2, the response is not acceptable. The authors should try to give a complete literature review on the subject. In addition, mentioning a short list for the important applications of E.coli is necessary.

Concerning Comment #4, the response is “… The paper aims to establish an innovative system and method for expressing human collagen, which does not involve methods of protein design …”, while in response to comment #2, the authors mentioned that “… in this article, we emphasize some advantages of E. coli as a system for the expression of heterologous proteins”!. These seem inconsistent argumentations for escaping from improvement of the manuscript.

Therefore, the manuscript cannot be recommended for publication (borderline to rejection), unless a significant improvement based on the original comments implemented by the authors.  

Comments on the Quality of English Language

Minor editing of English language required.

Author Response

Comments 1: the response of the authors is not acceptable. The mentioned reference is a review containing the important points regarding the collagen. Therefore, It should be addressed if they are discussing on the ECMs.

Response 1: Thank you for your advice. This reference ‘Prevascularized micro-/nano-sized spheroid/bead aggregates for vascular tissue engineering’ was recommended to be cited in the first sentence ‘Collagen, the principal structural protein of the extracellular matrix (ECM), plays an important role in preserving tissue architecture and modulating cellular physiological environments’. This review summarized advanced biofabrication methods in microvascular engineering, including extrusion-based and droplet-based bioprinting, Kenzan, and biogripper approaches. Also, the paper discusses the interaction between ECM and GF (growth factor) after the aforementioned methods have been applied. One case of applying collagen-glycosaminoglycan scaffolds to microvascular tissue engineering was deeply analyzed. Although this article involves the application of collagen in tissue engineering, the sentence in the text merely states the importance of collagen without delving into its applications. So, this reference will be not cited.

Comments 2: the response is not acceptable. The authors should try to give a complete literature review on the subject. In addition, mentioning a short list for the important applications of E. coli is necessary.

Response 2: E. coli has a wide range of applications in many fields, including virology, production of organic chemicals, biosynthesis of bulk chemicals, production of natural products, Environmental remediation, carbon dioxide fixation,genome editing and so on, our research focuses on the use of E. coli as protein expression systems to produce recombinant collagen. We cannot possibly enumerate all the applications of E. coli, so we have chosen the references that are most consistent with protein expression. We believe that this focus can make our research more attractive to target readers and help to highlight the novelty and importance of our research. So, the second reference will be not cited.

Comments 4: the response is ‘ The paper aims to establish an innovative system and method for expressing human collagen, which does not involve methods of protein design’, while in response to comment #2, the authors mentioned that ‘in this article, we emphasize some advantages of E. coli as a system for the expression of heterologous proteins’! These seem inconsistent argumentations for escaping from improvement of the manuscript.

Response 4: Our research focuses on the development of an innovative system and method for the expression of recombinant collagen, in particular the hydroxylation of collagen by incorporating proline and hydroxyproline in E. coli. Our goal is to provide a clear research focus, rather than introducing areas that are not directly related to the discussion, which may distract readers ' attention. We believe that our papers will be more direct and influential by maintaining this focus. We hope that these responses will address your concerns and further clarify our position. We appreciate your time and effort to help improve the quality of our papers. Although this manuscript does not involve the computational design of recombinant collagen, one manuscript we are preparing is about the molecular design of collagen. In this new manuscript, a considerable amount of space will be dedicated to the design of recombinant proteins, and we plan to cite the closely related literatures you recommended.

Reviewer 3 Report

Comments and Suggestions for Authors

Dear Authors,

The manuscript seems to be improved after revising as per suggestions. I think the revised MS can be accepted for publication.

Author Response

Thank you very much for your positive comments about our previous submitted paper-" Preparation and characterization of hydroxylated recombinant collagen by incorporating proline and hydroxyproline in proline-deficient Escherichia coli" in the bioengineering-3183892.